# Changing attitudes towards female genital mutilation. From conflicts of loyalty to reconciliation with self and the community: The role of emotion regulation

Afi Agboli[1,2]*, Fabienne Richard[3,4], Mylene Botbol-Baum[1,5], Jean-Luc Brackelaire[1,6,7], Annalisa D'Aguanno[4], Khadidiatou Diallo[4], Moïra Mikolajczak[1,8], Elise Ricadat[9,10], Isabelle Aujoulat[1,2]

**1** Institute of Health and Society (IRSS), Brussels, Belgium, **2** Université Catholique de Louvain, Brussels, Belgium, **3** School of Public Health, Université Libre de Bruxelles (ULB), Brussels, Belgium, **4** *Groupe pour l'Abolition des Mutilations Sexuelles* (GAMS-Belgium), Brussels, Belgium, **5** Health, Economics, Ethics, Law, and Social Issues (HELESI), Brussels, Belgium, **6** Institute for the Analysis of Change in Contemporary and Historical Societies (IACS), Louvain-La-Neuve, Belgium, **7** Faculty of Law, Université de Namur (UNamur), Namur, Belgium, **8** Psychological Sciences Research Institute (IPSY), Louvain-La-Neuve, Belgium, **9** *Centre de Recherche Psychanalyse*, *Médecine et Société* (CRPMS), Université de Paris, Paris, France, **10** Institut de Recherche Saint-Louis, Paris, France

* afi.agboli@uclouvain.be, afisophieagboli@gmail.com

**Data Availability Statement:** All relevant data are within the paper. This is a qualitative study on a

## Abstract

The practice of female genital mutilation (FGM) is a social norm embedded in the patriarchal system and is resistant to change due to its roots in the tradition of the practising communities. Despite this difficulty in change, some women succeed in changing their attitudes towards the practice. In trying to understand what makes these women change their attitudes, we identified in a previous study, the critical life events at which change occurs (*turning point*). These *turning points* were described with emotions and conflicting feelings based on which we hypothesised that emotion regulation and the resolution of conflicts of loyalty might be possible mechanisms that explain the change of attitudes by the women. In this article, we sought to investigate how the mechanisms interact and how they were at play to explain the change. We, therefore, triangulated our previous data, fifteen women interviewed twice, with the published life stories and public testimonies of 10 women with FGM, and interviews of six experts chosen for their complementary fields of expertise to discuss the emerging concepts and theory, generated by our study. The data were analysed using framework analysis and an element of the grounded theory approach (constant comparison). As a result of our theorisation process, we propose a model of change in five stages (*Emotion suppression*, *The awakening*, *The clash*, *Re-appropriation of self*, and *Reconciliation*). This describes the process of a woman's journey from compliance with FGM and community norms to non-compliance. Our study reveals how the women whose stories were analysed, moved from being full members of their community at the cost of suppressing their emotions and denying their selves, to becoming their whole selves while symbolically remaining members of their communities through the forgiveness of their mothers.

highly sensitive topic, based on the life stories of 15 women, who agreed to participate provided their testimonies would remain strictly confidential and may not be shared beyond the members of the steering committee involved in the analysis with the main researcher. Although all the transcripts have been anonymised, the women participants, who live in the same country (Belgium), are part of a small and highly connected community. They have unique experiences that may make them identifiable, should the transcripts of their interviews circulate. The data within the paper, comprised of all anonymised excerpts include all the data necessary to support the study's findings. Therefore, the full transcripts cannot be made available by request to other researchers worldwide. This is in line with the ethical concerns for the participants' confidentiality expressed in our research protocol, as it was approved by the Ethics Committee (Comité d'Ethique Hospitalo-facultaire) of UCLouvain with the reference number: 2013/21NOV/522 dated10th July 2017.

**Funding:** The author(s) received no specific funding for this work.

**Competing interests:** The authors have declared that no competing interests exist.

## Background

Female Genital Mutilation (FGM) is a public health concern with harmful consequences on women's health [1, 2]. FGM is gender-based violence that violates the bodily integrity of women and girls. The practice which is an intentional partial or total removal of female external genitals [1, 3, 4], contributes to the cultural identity of the woman who receives a mark on her body. This mark is seen as perfection of physical appearance within the culture [5]. Elderly women, guardians of the tradition, always strive to ensure that the practice is passed on from generation to generation. Consequently, the fear of seeing other members of the community performing the practice on their daughters [6] results in women fleeing and migrating to western countries, at times in very difficult conditions, to protect their daughters [6–8].

Many health consequences have been documented to result from FGM. Among them, are physical consequences [3, 9], and psychological consequences which include anxiety, Post-Traumatic Stress Disorder, and depression [10–13]. The practice of FGM is a social norm held in place by individual attitudes of parents and other family members' decisions of performing FGM on girls [2]. The cultural values imparted [14] and other related norms embedded in the patriarchal system together with the associated meanings of performing FGM make the practice resistant to change [8]. Despite the practice of FGM remaining a social norm that is difficult to change because of its deep roots in the tradition of the communities, some migrant women succeed in changing their attitudes towards it and do not perpetuate the practice on their daughters. Attitudes change after migration regarding FGM have been reported in other European countries [5, 15, 16]. Changes in the social norm towards FGM among migrants as a result of living in settings where FGM is not the norm have also been evidenced [5, 17–20].

The migration context appears to be a favourable platform for the change of views towards the practice of FGM. It leads individuals to reflect on the traditional practices of their countries of origin [21]. Factors participating in creating an environment that favour the change have been documented. The environmental factors in contexts where the practice is against the law influence the change of attitudes among migrant people [15, 22, 23]. The criminalisation of FGM in Western countries is seen to be an important step towards saving girls from the harmful consequences of FGM [24]. New environments, the migration journey coupled with anti-FGM interventions, and the encounter with health services have been documented as factors contributing to significant shifts in the women's attitudes towards FGM as they reject the practice and tradition [8, 17, 25–27]. Reduced social pressure from extended families was noted because of the distance factor in the context of migration [25], and the length of time spent in host countries [18, 27, 28] also plays a role. Women from the same religious background make them understand that FGM is not a religious principle [16].

Scholars have studied the change of attitudes towards the social norm in the country of origin and, using existing models of change at the community level have been applied [13, 22, 29–31]. However, we are not aware of any model of change at the individual level in the country of origin. The Transtheoretical Model (TTM) or stages of change model as developed by Prochaska and DiClemente [32] has been applied to reflect behaviour change and conceptualised the change of FGM/C in terms of stages of change, but the exact identifiable stages could not be defined [13, 30, 31]. Another model of change at the community level is the social convention theory developed by Mackie and LeJeune [22] who stressed that as the decision to cut depends on the wider community, a motivated mass of people collectively, deciding to make to abandon the practice publicly might end FGM as it was the case of foot-binding in China [33]. There are challenges in changing attitudes towards FGM and all these approaches and models applied have limitations as far as FGM is concerned, and the impact of interventions aimed at changing attitudes towards FGM was mostly studied at the community level. How

changes occur at the individual level remains an under-investigated issue, which this present study aims to address. To date, the different approaches discussed in this section have not led to a reduction of the practice as such due to the insufficiency of strategies that work.

In a previous study [8], we identified and described specific significant events *(turning points)* in the life trajectory of the women that were associated with a change in attitude towards FGM in a migration context. These *turning points* were related to encounters with health professionals, education, social interactions with other cultures and their own culture, experiences of motherhood, repeated pain during sexual or reproductive activity, and witnessing the effects of some harmful consequences of FGM on loved ones. They were defined as significant events including social interactions that challenged the women's norms and any other expectations associated with the practice of FGM, leading to changing their attitudes towards it, and taking action to quit their community to protect themselves and their daughters from the practice. With these *turning points*, we had hypothesised that emotion regulation and the resolution of conflicts of loyalty might be two common mechanisms involved in the empowerment of the women as they decided to act against FGM for themselves and their daughters. This paper seeks to further explore how these two processes may interact to explain the change in their attitudes towards traditional norms.

## Methods

A qualitative methodology informed by the grounded theory approach was used to allow us to generate the hypotheses mentioned above [8]. The approach inductively helped to consolidate these hypotheses through constant comparison [34], taking into account the narratives of the women who had participated in the first study (n = 15) and comparing them with published testimonies (n = 10), while further exploring the literature and consulting experts including one expert with lived experience to discuss and consolidate our emerging theorisation.

### Data sources

In addition to the continuous exploration of the literature, three sources of data were used to help us theorise the process of change experienced by women as they turned their backs on the practice of FGM and took action to protect future generations from it: (a) Fifteen women were interviewed twice in our previous study [8]. We chose the method of biographical narrative interviewing method (BNIM) as developed by Wengraf [35] in order to produce narratives relating to life events. In the BNIM approach to data collection, the interviewee is seen in two phases and sometimes three (not always present), with the first interview being unstructured and the consecutive interviews building on the previously collected data [35]. Thus, we had 30 transcripts derived from the interviews of the 15 women participants in this study who had all undergone FGM in their countries of origin during childhood. They all originated from sub-Saharan African countries and were living in Belgium at the time of the interview. Full details of the sample and the description of the process of the interviews can be found in our publication [8]. (b) In addition to the 30 original interview transcripts, we included the life stories and public testimonies of 10 women referred to as 'norm leaders' (see Table 1). (c) The third source of our data comes from interviews we conducted with six experts, including an expert with lived experience, to discuss the emerging concepts and theory generated by our study. As complex intrapersonal psychological processes were involved in the hypotheses generated by our previous study [8], we aimed at recruiting psychologists with complementary fields of expertise, that we felt were relevant to inform our emerging theory: sexual health and identity issues (Elise Ricadat), emotion competence and regulation (Moïra Mikolajczak), transcultural psychology (Jean-Luc Brackelaire), clinical psychology involved in FGM follow-up and specialised

**Table 1. Table of the books of norm leaders.**

| Names of NL | Title of the book | Year of publication | Country of origin | Host country |
|---|---|---|---|---|
| Thiam, A | La parole aux négresses [The voice belongs to the negress] | 1978 | Senegal | France |
| Diallo, K | Mon jardin dévasté [my devastated garden] | 1991 | Senegal | Belgium |
| Dirie, W. & Miller C. | The desert flower | 1999 | Somalia | Austria |
| Barry, M | La petite peule [The little polar] | 2000 | Senegal | France |
| Abdi, N | Larmes de sable [Tears of sand] | 2003 | Somalia | Germany |
| Koita, K | Mutilée [Mutilated] | 2005 | Senegal | France |
| Bah, D | On m'a volé mon enfance [My childhood has been stolen] | 2009 | Guinea Conakry | France |
| Miré, S | The girl with three legs | 2011 | Somalia | USA |
| Kanko, A | Parce que je suis une fille [Because you are a girl] | 2014 | Burkina Faso | Belgium |
| Bowin, L | Swimming in the red sea | 2018 | Guinea Conakry | Canada |

in traumatic memory (Annalisa D'Aguanno). In addition to these various specialties in psychology, we also discussed with a philosopher, specialised in bioethics, norms, and the capabilities approach background (Mylene Botbol-Baum), and an anthropologist who accepted in the first place, but was later unfortunately not available anymore. Last but not least, it was of great importance for us to have the lived experience as an expert who had experienced FGM herself (Kadhiatou Diallo). The experts were approached personally by the first or last author, who knew them for their specific fields of expertise, which did not include FGM-related knowledge or experience for four out of six of them. The first author and the last author discussed the purpose of the study with them and met them individually through videoconference due to the Covid-related restrictions. The consultation with the expert with lived experience was done face-to-face by the first author alone. The discussions and exchanges ranged from 45 minutes to one hour with an average of 50 minutes. Notes were taken during the discussions. Prior to the meeting, the experts had had the opportunity to read and comment on a draft document that summarised the preliminary results on the applicability of the concepts of emotion regulation and the resolution of conflicts of loyalty to understand the process of changes in attitudes towards the norms associated with FGM. Every expert was sent a synthesis of the main theoretical issues arising from their interview, to complete and validate. During the process of further integrating the different theoretical perspectives to answer our research question, the experts had the opportunity to further contribute by commenting on several draft versions of this article. Eventually, all were invited to co-author the article for their contribution.

## Ethical considerations

The first source of data: The women from the first data source were recruited from our previous study [8]. The study received approval from the Ethics Committee *(Comité d'Ethique Hospitalo-facultaire)* of Saint Luc University Hospital Brussels with reference number: 2013/21NOV/522 dated10th July 2017.

Second data source: the books are published testimonies of the life stories of women who strongly oppose the practice of FGM. No consent was needed for this data source.

Third data source: The experts. All the experts agreed to take part in the study and verbal consent was received from them. Before meeting the experts, we sent an email to each expert to describe the objectives of the study and invite them to share their expertise in the process of our emerging theorisation regarding the mechanisms that explain the change of attitudes towards the practice of FGM. They agreed on having their names and their expertise to be reported in the article and they agreed to co-author the article.

## Data analysis

The data analysis occurred in three steps following a sequential design and through constant comparison.

**Secondary analysis of the 30 transcripts from the first study.** The objective for this first step was to systematically screen the 30 transcripts from the first study again, which had initially been inductively analysed to reveal specific *turning points* in the lives of the participants. In this secondary analysis, we aimed to examine how the emerging categories of emotion regulation and conflict of loyalty resolution were represented in the data. The *turning points* revealed in the first study constituted a starting list of categories for the secondary analysis of the transcripts. More precisely, we verified that the *turning points* were indeed associated with experiences and the expression of emotions as well as conflicts of loyalty, as we had hypothesised at the end of our first study. The data were synthesised using a framework analysis approach, a process that involves systematic indexing of responses in line with a given framework [36]. This was based on the dimensions of Gross' model of emotion regulation [37] on the one hand and on the model of the processes of informing loyalty on the other [38].

The two theories referred to for this framework analysis may be briefly presented as follows:

*Emotional regulation.* Emotion regulation is a key contributor to social functioning and refers to the processes by which individuals influence what emotions they have when they have them, and how they experience and express them [37]. Such control processes may be conscious or automatic [37, 39] when responding to environmental demands and may take effect at one or more points in the emotion generative process. An umbrella term for the emotional state which is *'affect'* is viewed by Gross [37], as stress responses, emotions such as anger, sadness, and mood such as feeling, or down. Affect refers to an individual's life and beliefs that preclude the understanding of social life as a system of rules and norms [40]. Gross [37] develops a process model of emotion regulation which allows for the identification of when regulation may be understood to have occurred. The model has five major points of focus (*situation selection*, *situation modification*, *attention deployment*, *appraisal*, and *response modulation*). Emotion regulation is goal oriented. Goal oriented emotion regulation strategies related to bodily expression include emotion suppression, in which people actively try to inhibit their emotional expression [37]. Expressive suppression is defined as the inhibition of emotional expression, such that an outside observer would be unaware of an individual's internal emotional experience [37]. The suppression does not always lead to the desired changes in emotional experience but can lead to a negative emotional experience instead.

*Conflicts of loyalty.* Loyalty is a feeling or an attitude of devoted attachment, even where it may not be deserved [38]. Loyalty conflicts occur when a person feels trapped in a disagreement between two people, and they expect the trapped person to support them over the other. The literature describes mainly children's perceptions of one parent's pressure to side against the other and the conflict is expressed by the feeling of being trapped, torn, or 'caught between parents' [38, 41]. Research on children in foster care presents a model of processes informing loyalty to help to understand the experience of children in foster care and how they negotiate living with a new family while retaining their place in their birth family [38]. The model that emerged from their results encompassed seven theoretical categories. These categories enabled different relationships children have with their parents' loyalty to be distinguished: *new realities; considering position; making sense; relating emotionally; working out loyalties; considering others' perspectives; and self-determination.*

Loyalty conflicts have generally been the focus of attention in the literature on children of divorced parents [41, 42], as well as in foster care literature [38]. It has not been applied to

adults much; however, the concept seems essential in our study because the women felt trapped between what the community expects from them and what they want for their daughters.

*Original analysis of the books (n = 10).* All 10 books were read by the first and second author to identify the *turning points* in the women's lives that had prompted them to change their views and attitudes toward the practice of FGM, and other related norms. These were compared to the *turning points* drawn from our previous study [8]. Then the two theoretical frameworks mentioned above were applied by the first author to systematically screen for expressions of emotions and conflicts of loyalty resolutions with those *turning points*.

*Consultations with experts.* In the third stage, after having confirmed that aspects of emotion regulation and conflicts of loyalty resolution were indeed present in the women's narratives as they reported *turning points* in their lives, we submitted our results to the different experts included in our study to discuss our emerging theorisation, and to help us refine it, by collaboratively working out the interrelations between the two hypothesised mechanisms of change, i.e. emotion regulation and conflicts of loyalty resolution.

The notes taken during the different discussions were compared and analysed by the first and last authors to further clarify our emerging theorisation, which was also discussed with co-author FR on several occasions and enriched through a review of some published material suggested by the experts. For instance, to understand trauma and its impacts on the body, we read the literature of Van der Kolk [43]; Rothschild [44]; Salmona [45], and Hoschild [46]. To become more familiar with narratives and the capabilities approach, we read the books of Botbol-Baum [47] on '*Bioethics dans les pays du Sud*' [Bioethics in the southern countries] and Nussbaum [48] on women and human development, the capabilities approach. We iteratively pursued this theorising and integrative process, until we came up with a model with five stages (*Emotion suppression*, *The awakening*, *The clash*, *Reappropriation of self*, and *Reconciliation*), which we present hereafter in our results section, to explain the process of the change of attitudes in women with FGM as they move from being full members of their community at the cost of suppressing their emotions and denying their selves to becoming their whole selves while symbolically remaining members of their communities of origin. The results hereafter are presented according to the five stages of our proposed model, and integrate the findings from all three steps of our analysis.

## Results

### From suppression of emotions to reconciliation with self and community

The model that emerged from our findings consists of the five stages mentioned above. Each stage refers to the processes that were found to explain the women's attitudes towards change. In the diagram below (Fig 1), the five stages are depicted as parts of a linear process, but we are aware that these stages might not always be linear, as there may be overlap and a woman may move through and between stages of the model numerous times or simultaneously.

**Emotion suppression.** Our results show that women's emotions were often repressed and suppressed before the *turning points*. Emotions are socially constructed so culture has an impact on the expression of emotions. Girls are taught to be brave and not to cry during the procedure to prevent disgracing their families.

> "*Afterwards, we are told: you must remain courageous, never complain, you have passed the first stage, there will be others that will come with pain too, but do not forget that a good woman must always bear the pain, and must suffer.*" Interv_2

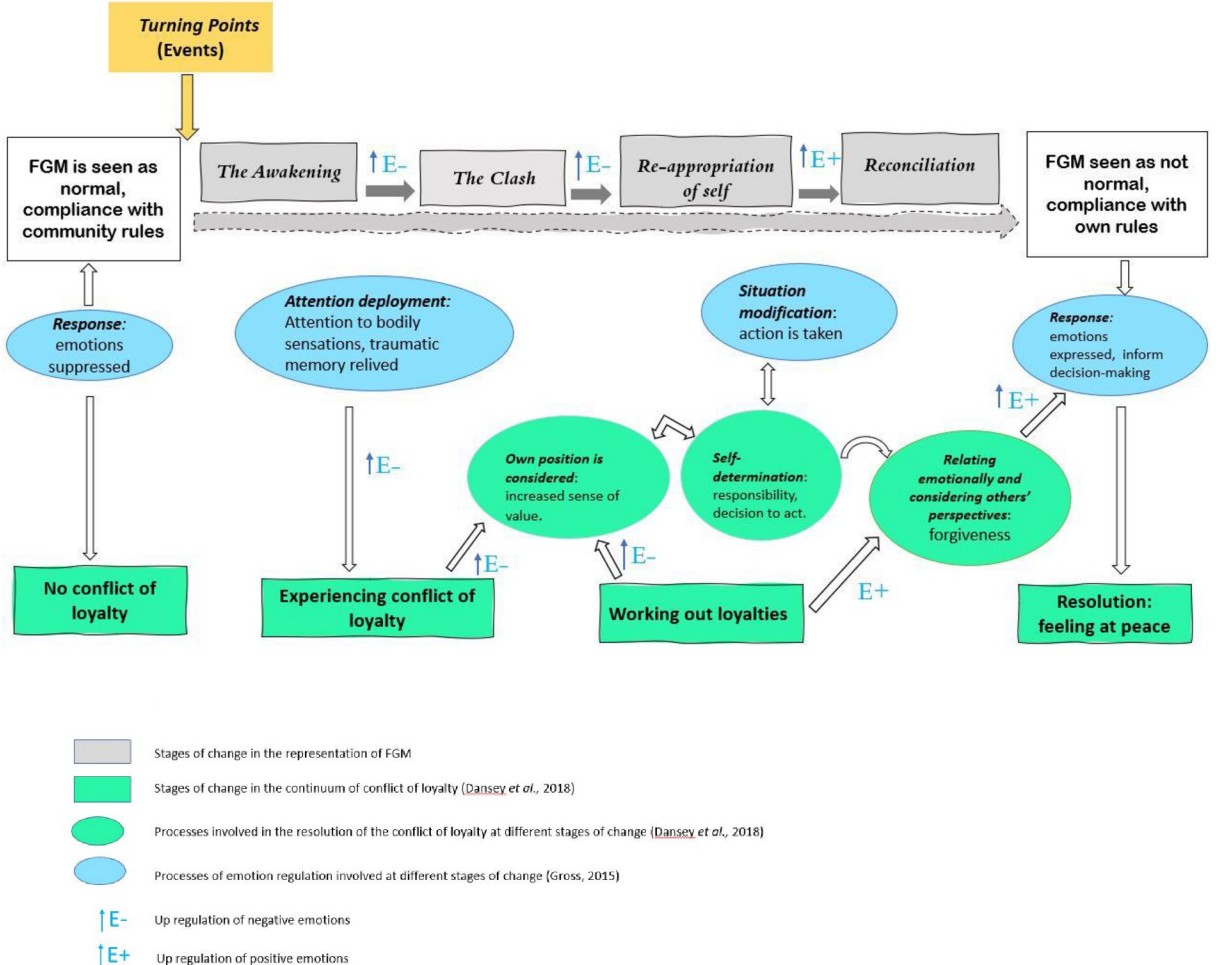

**Fig 1. Psychological processes involved in the stages of change.**

*". . . I got used to being in pain; not crying anymore but clenching my teeth. . .later on, I continued clenching my teeth as I was always told. . ."*

Kanko [49]

Before the occurrence of the *turning points*, the suppression of emotions was associated with the acceptance and internalisation of the practice of FGM. The women used to comply with the social norm as prescribed by the patriarchy. The patriarchal system determines and prescribes how individuals ought to behave regarding certain feelings (*affect*) and emotions. The cultural grounding of emotional expression is reflected in the whole procedure when at times, women are instructed to be brave and endure pain and suffering without complaint. This is a norm in the patriarchal system. Even though the women were not allowed to express any emotion, they felt them, and they were affected by them. Some women described their initiation stage as being taught about their gender roles by the older women who are the guardians of the tradition. They were guided by the instructions from the older women to endure pain and suffering as their accepted fate. Fear was what generally drove women to comply with

the social norm to avoid being ostracised and isolated from the community and have their daughters unmarried.

*"I was constantly in fear and felt the anger, but I did not know where they were coming from. Because something tells you why 'you'? And during the initiation phase, they talked about your recognition as part of the community, you are going to become a woman, you are going to serve the cakes, you are going to be pure and say good prayers but they will never tell you the intensity of the pain, just to bear it and not talk about it. It's like you are being plunged into the unknown which is unbearable and that's the problem."* Expert with lived experience.

The suppression of emotions which results in the inhibition of the emotional expression is one of the response components according to Gross' model [37]. It represents the last stage of Gross' model while it is the starting point of our model. Suppressing emotions ensures community membership and the women usually do as they are instructed to respect the elders. Silence is also imposed by the patriarchy and the women are trained socially to bear the pain at important moments of their lives (for instance, first sexual intercourse on the wedding night, as well as at childbirth and not screaming during labour).

*"The patriarchy uses violence to assign identity and thus, construction of cultural identity is through the bearing of pain without complaint, so involving suppression of emotions"* Expert in Psychology and Traumatic memory.

**The awakening.** At the second stage of our model, which we call *the awakening*, emotions are expressed with the *turning points* described in the narratives of the women who participated in our previous study, as well as those from the norm leaders whose testimonies we analysed.

*"The emotions that were expressed evolved with the process of emancipation and questioning of norms. These were 'affects' that were acknowledgement that precedes the change of norms. Turning points may be seen as a normative conversion, allowing the capability to act against one's s community beliefs, which is also a definition of emancipation. This conversion of self is indeed a turning point for it allows going beyond the conflicting emotions between community and self".* Expert in Philosophy and Bioethics.

At this stage, the emotions are felt and acknowledged. In Gross' model, the acknowledgement refers to the *attention deployment* stage, and to the events and emotions it generates. As they recalled the *turning points* in their lives, most women described sensations in their bodies in the form of pain in the womb and goosebumps. These prompted them to reflect on their own experiences of the procedure of FGM, which in turn awakened their traumatic memories. They described these *turning points* as events causing them to relive the procedure of FGM, thus experiencing emotions of fear, helplessness, and betrayal.

*"[. . .] then on the skype, my mother-in-law had called me. . .soon you will be with us, I have prepared everything, and the circumciser is ready to purify the little one like this, she will be clean and everything. Baf! It was like a slap! I felt something in my belly".* Interv_1

*"My stomach sank as I recalled the sound of scissors, the horror and the fear that overwhelmed me when I faced the men in white coat. . ."*

Miré [50]

We know from our previous study [8] that there might be different types of events leading to *turning points* and thus the awakening of emotions. In some narratives, an encounter with a health professional (a gynaecologist or psychologist) was key in provoking the awakening. The role of others when old memories are stirred up and emotions relived and expressed by the women, sometimes for the first time is crucial. Indeed, the women need to experience a sort of 'safety net' while their negative emotions are recalled as they reconsider their own experiences of the procedure of FGM.

At that stage, most of the women reported vivid emotions such as anger, fear, despair, and suffering, which they had been used to conceal not only from others but also from themselves. With external help, the traumatic nature of their experience would be acknowledged, thus allowing their emotions to be integrated as part of their experiences and selves, rather than suppressed again.

*"The dimension of trauma should be considered here as there is violence against women's bodily integrity. Also, in their traumatic experience, the women's sense of identity may have split due to their emotions and stress (traumatic cleavage). Being supported by a professional who is used to dealing with trauma when they relive it, could therefore be considered the starting point to building their new identity and a way to foster it in a personal and subjective way"* Expert in Psychology and Identity issues

An expert considered how the acknowledgement and expressions of negative emotions were crucial at that stage to help the women move forward in changing their attitudes towards the practice of FGM as they enabled the up regulation of these negative emotions. Negative emotions are not to be down regulated at this stage but up regulated instead.

*"The rising of negative emotions is important for women to acknowledge and accept them, and thus, the up-regulation of these negative emotions will push the women to do something and act"* Expert in Emotion Competence

The negative emotions relived by the women interviewed in our study as well as those whose published testimonies were analysed. These emotions were derived from the consequences of FGM and brought a lot of traumas into the women's lives. For example, trauma made one woman bleed in painful circumstances, even when unrelated to FGM.

*"Anything that upset me makes me bleed. . .I can't believe I am bleeding again, at hearing about the Brussels' attack."*

Bowin [51]

**The clash.** In the third step of our model, which we refer to as *the clash*, we looked at how the awakening and up regulation of negative emotions, that were associated with the acknowledgement of trauma led to inner conflicts of loyalty. Women experienced conflicts of loyalty when they wanted to protect their daughters as they did not want them to go through the same experiences. This gave them to have some mixed feelings as they became aware of different norms and values that would define them in different ways, in their own eyes and that of their community of origin. The feeling of being trapped between the two constituted a conflict of loyalty. As they experienced such conflicts of loyalty, the women became further aware of their bodily sensations and emotions such as embarrassment, shock, fear, etc.

*"I was constantly in fear and felt the anger, but I did not know where they were coming from [. . .] your own community has fooled you; you feel bad because you want to be a good woman according to the religious norms of the community". Expert with Lived Experience*

At this stage, the negative emotions are still present. This makes the women try to come out of that state. However, the conflict of loyalty generates in turn new sufferings and negative emotions and needs to be solved to gain peace. According to Dansey *et al*.'s model informing loyalty [38]. *Considering one's own position* is a key stage to moving past this when trapped in a conflict of loyalty. In our study, the women at this stage would consider their positions as mothers and the devoted attachment to both their daughters and their communities would conflict, thus prompting them to work out how to prioritise those loyalties. This was the stage where a need was perceived to weigh up and choose, to emerge from the conflicts they were experiencing. Indeed, choosing to protect their daughters by not perpetuating the practice would isolate and even ostracise them from their communities of origin. The women, therefore, reported experiencing conflicting representations of motherhood: what made them believe they were good mothers in their own eyes and that of the society in their host country by protecting their daughters from the practice of FGM, would make them bad and irresponsible mothers instead in the eyes of their own mothers and their communities of origin. Deciding to protect their daughters, meant the death of their affiliation with their original communities because they were defying the prescribed norms established by the patriarchal system.

*"I think of my daughter. . .but I want to be a good mother in my community also » Interv_15*

*". . .I was convinced that I was born free, not with a pot in one hand and a broom in the other. . .I couldn't understand how a hand that caressed you, wiped your tears, reassured you, comforted you, could hurt you. . .I was confused. . .but I was committed to my sister and brothers, not to abandon them."*

Barry [52]

Along with conflicts of loyalty, conflict of interest might be experienced, as women may be trapped in the feeling of losing the status that they gained through the procedure of FGM. Moreover, when women oppose a norm, their opposition becomes an element that disturbs and endangers the patriarchal system as a whole.

**Re-appropriation of self.** The next step in our model which we named *Re-appropriation of self* suggests that as they move forward, the women need to actively engage in processes of *Re-appropriation of self* whereby they accept the need to reconsider their positions and see and value themselves for what they are. This is different then, from what is valued in their communities of origin. This is what Dansey *et al*. [38] call the recognition of a new position and re-conceptualisation of a sense of value.

*Re-appropriation of self* is an important stage in the women's journey. It entails their identity re-appropriation as well as that of their bodies and sexuality. The women explored their responsibility to themselves and sought to re-appropriate their identity and rebuild their self-esteem and self-confidence. Some women became aware of their inner internal agency and were able to identify and express personal qualities such as being curious and stubborn in response to the patriarchal system that sees women as subordinates. Acknowledging such personal characteristics enabled them to preserve their sense of self. One woman reported:

*"I'm a bit stubborn, there are things I didn't accept. For example, I couldn't stand the veil before the wedding and after the wedding, I wore it in my own way, that's me." (Interv_4)*

This stage also entails women starting to think by and for themselves to discern their identity as distinct from the one imposed by the community. At this stage, the participants in our study as well as the women who shared their testimonies through published material started to value what they wanted for themselves instead of what the community wanted for and from them. There was an awareness of their internal drive to claim back what they had lost and therefore, their bodies could be reclaimed. The women recognised that some aspects of their future were within their control by wanting to undergo the reconstruction with regards to the lost physical part of themselves (the clitoris) so as to regain their sexuality. The control over their own lives placed more responsibilities on them to make changes to improve their lives. This was a shift from the powerlessness they felt before the *turning points* and while they were dealing with conflicts of loyalty. As a woman participant reported:

*"I had wanted to gain something but rather I lost something instead and that they never told me I felt betrayed. Now, I want it back, I want to do the reconstruction of the clitoris" Interv_3*

The issue of responsibility is an important part of the journey at this stage. Accepting responsibility for themselves and making their own decisions for their wellbeing and that of their daughters gave them control over their own lives. The women's sense of self-worth increased as the positive emotions were released at this stage of resolving the conflict of loyalty.

*"The re-appropriation of the body and sexuality implies the responsibility towards oneself and others; responsibility and identity are articulated. This re-appropriation seeks to (re)define and (re)delimit not only identity but also the roles between oneself and others; that is to say the power and the competencies belonging to each, entering into the transaction by allocating responsibilities with their duties with respect to others" Expert in Transcultural Psychology.*

In addition to the help or advice received from professionals, those women who had a partner from a different cultural background stated how they had been helped by their partner to understand that sexual intercourse ought not to be painful and that they had a right to be satisfied with their sexual lives. This is an important role played by the others in the women's change of attitudes.

*"The fact that my partner supports me, encourages me, and has offered to help me with the reconstruction, and that he will pay the costs has also given me strength to move forward and I feel valued" Interv_12*

The women acknowledged the viewpoints of the professionals and partners but made their own decisions, which signals the phase of *Considering others' perspectives* in Dansey *et al.*'s model as important for influencing individuals' positions and loyalties. In Gross' model, this stage refers to the *Situation modification* which in our study was materialised by actions such as fleeing with their daughters, going forward with clitoris reconstruction or defibulation or writing a book (becoming an activist). They made sense of their situation by realising that they did not have to suffer without saying anything. Women also reckoned that the assistance from the safety net of helping agencies (for example the non-profit organisations and some health professionals such as psychologists), was important in this phase of dealing with their emotions of fear, anger, and the accompanying traumas. This was an empowering experience for the women, as they saw these relationships as contributing a positive contribution to their wellbeing. Some positive emotions were brought up when the women felt supported by significant others.

At this stage of their journey, *Self-determination* was established by the choice the women made in another context which enabled them to prevent further FGM. The women reported their ability to be assertive and make plans for themselves for the future, for their comfort, and for independent lives. This encouraged them to have positive feelings about themselves and think positively about their future. They also made plans for their relationships with their children, families, and new romantic partners.

*"You see, I have plans, a lot of plans for the future, first of all, I tell myself that, I'm going to move to France so that I'll be close to my parents and then do something else, like going back to school and furthering my education" Interv_4*

**Reconciliation.** In the last stage of our model, women have managed to not only reappropriate themselves and their bodies but also to reconcile themselves, even if sometimes just symbolically, with their mothers and communities. At this stage, the women expressed being able to identify with their mothers, understood that they could not have acted differently, and ultimately forgave them for what they were once subjected to. They did not apportion any blame to their mothers.

*"And when I think that my grandmother certainly underwent this in a more archaic way . . . I say to myself that we need to understand them. . ." Interv_9.*

At this stage, the empathy the women felt towards their mothers enabled them to overcome their sense of blame towards them. They became able to *relate emotionally* again to their mothers. Many participants reported their struggle to understand why their mothers had them undergo FGM. But they also thought of the fact that their mothers had gone through the procedure in an even more violent way than they had. Some women reported anger towards their mothers, but later did not want to blame them.

At last, the women at this stage report being able to forgive their mothers for what they have once been subjected to, as they acknowledge their mothers did probably not have a choice to act differently because of the social pressure inherent to the context and time at the moment. By contrast, they see themselves as having a choice, especially in the context of their Western host countries where the practice of FGM is prohibited by law.

Forgiveness towards their mothers led to a symbolic reconciliation with the community. Some women participants depicted a transformation in their perspectives through their acceptance and forgiveness of themselves and their mothers. By forgiving their mother, the women did not condone their mothers' actions but rather they freed themselves from the anger and the negative feelings that had threatened their wellbeing. As one woman put it: "*I love my mother and my people*" Interv_13. They understood that forgiving their mothers was part of the process that would enable them to move forward and be part of the community again, even if just symbolically. They let go of anger and blame towards their mothers.

*"Letting go allowed me to live my life so that I don't live other people's lives. Then, I gradually learned to regain confidence in my parents. I undertook not to judge my mother anymore. . .But I wouldn't want her story to be really mine, nor that of my children"*

Kanko [49]

*"Our mothers, my grandmother, are the bearers of this tradition to please the men but it is ignorance. And when you realise that, your anger fades away, you just know that you can't be*

*like them. . .I have to protect my daughter and other little girls who undergo the practice during the school holidays. I am less angry today"*

Bah [53]

At this stage, by seeing themselves as different from their mothers yet being able to identify with them and forgive them, the women demonstrate to have developed a *Self-determination* capacity despite their powerless position and to be able to successfully *Work out their conflicting loyalties* as Dansey *et al*.'s model [38] suggests.

*"I first began to reconcile myself with the part of my body that was ignored, and then with my community so as not to blame anyone"*

Koita [54]

## Discussion

The main aim of this study was to generate a better understanding of the mechanisms and factors that might explain the change of attitudes towards the practice of FGM in the context of migration with *turning points* that occur in the life trajectories of the women who challenge the practice of FGM. Emotions such as fear and helplessness were reported in our study as relating to the patriarchal ideals of what a normal and good mother should be. Negative emotions tend to be suppressed before the occurrence of *turning points*, as according to patriarchal ideals, women should endure pain and suffering without complaining. Therefore, anything that the women felt but were not able to express because there was nobody to hear them, tended to transform into the fear that would hinder their emancipation and freedom. Conflicts of loyalty inevitably arise when the women question the legitimacy of the rules and norms of their own communities. In our study, the women reported having felt torn apart by conflicts of loyalty as they sought to be themselves while being part of the communities. They challenged the prescribed norm when the lives of their daughters were at stake, which made them think back on their own experiences and feel the urge to protect them.

Our findings illustrate that as they were socialised in a patriarchal system, the women were not permitted to express their emotions, especially pain in three crucial moments of their lives i.e. during the procedure, on the wedding night, and during childbirth. These findings confirm other research findings for example Abdalla [55]; Malmstrom [56]; Fisaha [57]; and Morgan [58] in that, women were not allowed to express their suffering verbally during the procedure and in the social interaction of their everyday lives. Suffering and pain were understood as '*normal*' and as expected parts of a woman's life. This socialisation-based learning includes acts that demand silence without complaint. Verbal complaints were understood as a failure and also as shameful since these acts were connected to sexuality and reproduction. Silence and secrecy guard against the making of the feminine self [59], while the painful experiences are continuously engraved on the inside of the body. For them, being silent or silenced about one's trauma signifies the loss of power and self [60].

The women acknowledged that the FGM experience creates psychological trauma and is a potential cause of Post-Traumatic Stress Disorder (PTSD) for the women [11]. Trauma is considered a psychophysical experience where the traumatic event has a damaging impact on the mind and the body, and the bodily injury can result in fear of annihilation [44]. The results of this trauma invade the consciousness (flashbacks, sensory illusions, nightmares) and mean all

or part of the trauma is relived identically, with the same distress, the same terror, and the same physiological, psychosomatic, and psychological reactions as those experienced during the violent event itself [44]. Van der Kolk [43] asserts that the re-experiencing in the present moment of the physical sensations of past traumatic events triggers the experience of uncontrollable intense feelings. The impact of PTSD on the mind and body has been discussed by Rothschild [44]. She examines a normal response to that of a PTSD response and concludes that the most severe consequences of PTSD result from *dissociation*. Women who have undergone FGM manifest PTSD as well as trauma-related complications, shutdown dissociation, depression, and anxiety [10, 61, 62].

In the literature, suppression of negative emotions is thought to lead to poor long-term health outcomes [63, 64]. In our results, the rising of the negative emotions in women triggered sensations in their bodies, causing them to relive their personal experience of the procedure of FGM. These personal experiences brought some negative emotions to the surface which led to feelings of vulnerability in the women and the awakening of the traumatic memory. In theory, the negative emotions are to be down regulated by positive emotions to minimise the consequences of emotional experiences [65, 66]. However, the up regulation of negative emotions was key in fasting the women to move forward, as they started to question the norm in the process of becoming aware of the responsibility they have towards their daughters' safety as well as to their personal lives. As the traumatic memory was awakened, the women in our study became more aware of their own negative emotions (fear, betrayal, helplessness, pain, and a sense of being overwhelmed), and this triggered sensations in their bodies. The acknowledgement and acceptance of these emotions, called *the Attention deployment* in Gross' model [37], represents the explanatory mechanism of up regulation of negative emotions in *The awakening* stage of our proposed model. It also led the women to move through and reach the next stage, *The clash* (conflict) stage even though this was uncomfortable for them. That is the stage when most of them took action and fled from their countries of origin meaning that they were able to step out of the normativity of the social community. Thus, the awakening of their emotions fostered their possibility to take agency of their suffering body over the social expectations. This is an articulation of individual experience and the capacity of modifying a social norm. In the context of migration, these women were exposed to a variety of norms and were affected by conflicting norms that allowed them to make a choice to be agents and questioned the norm. The choice offered in the new context is one their mothers and grandmothers did not have because they were only participating in the patriarchal model which was the only narrative available to them. However, making a choice and acting on it implies the capacity to rewrite alternative narratives and also implies the concrete realisation of freedom as a transgression of their own norms while they respect the law of the host country [67]. The choice was also an important part of *Working out their loyalties* [38], which enable the women in our study to be able and determined to choose their daughters and take control. Dansey *et al.* [38] stress that this *Self-determination* is understood as being underpinned by empowerment. They created their own sense of meaning and acted to make their own decisions for themselves and their daughters.

## The body as a vehicle for empowerment and capability

The changes in their sense of self the women described in our study while they re-appropriated their bodies and sense of identity and responsibility towards themselves and their daughters, constitute a gain of empowerment. This is consistent with other studies on violence against women where women experienced changes in themselves through a change in their power and their concrete actions to take control over their own lives [68, 69]. This process includes

recognition of their power to make choices about their own lives and take steps to embrace that power. Fahs and Swanks [70] suggest that embodied resistance is when the body is used to contradict cultural norms; for example, the women resisted and decided to go ahead with the reconstruction of the clitoris or de-infibulation. Their bodies gave them the means to act on their decisions about themselves. This is a *capability* according to Nusbaum [48]. These parts of their bodies appeared meaningful to them, and they thought of protecting their daughters' bodies as well. Before the occurrence of *turning points*, what the norm imposed was contrary to what they felt. After the *turning points* had occurred, they were able to disclose their feelings and express their emotions. This was consistent with previous studies [8]. Under the practice of FGM, there was a sense of communal body. However, after the *turning points* had occurred, the women gained a sense of a more individualistic body, where their rights are promoted in terms of their decisions concerning their bodies. Their emancipation or empowerment is therefore seen as arising from an individual body towards a community, a bottom-up process. The narratives of the women enable them to gain the strength of internal resistance that comes from self-confidence by becoming responsible for the sequence of events experienced [47]. Furthermore, the capability to preserve oneself as a person within the community and act against one's own community beliefs represent the definition of concrete freedom [47].

A sense of agency and responsibility to make their own decision and to protect their daughters was a source of relief that acted as a counterpoint to their memories and pain. This is consistent with the findings of Koukoui *et al.* [71] that mothers were appreciative of the opportunity to raise their daughters in an environment that offered protection against FGM and took comfort in knowing that their daughters would know another destiny.

## The role of significant others in this process

The significant others played an important role in the change of attitudes of the women. Migration in the host country contributed to raising the awareness of differences between cultures and not being defined exclusively by FGM. The safety net of helping agencies and the assistance of significant others and health professionals were determining factors in the stage of resolution of conflicts of loyalty. They can help to identify women in all stages of the process to assist with their needs at that stage.

When they had partners from another culture, this was also significant because the women understood that one could be a member of a community without being cut. They no longer believed that women must endure pain and suffering when their partners made them aware of the possibility of living their sexual life openly without pain during sexual intercourse which was a clear difference from the expectations of their own culture. In the host country, the different interactions with others also helped in the construction of their identities. Identities are relational and socially constructed [72, 73].

Women were able to shift and exercise their internal agency as an empowering means to achieve the stages of re-appropriation and reconciliation. This demonstrates their direction in moving forwards which reinforces the women's self-esteem, security, and freedom. This shift also appears to enhance their determination to move through all the stages and to advocate against the practice.

## Forgiveness and empathy in the process of changing attitudes towards FGM

The women represent themselves within their communities by identifying themselves with their mothers. They can be themselves and, also, they can be part of the community. The identification is a form of empathy they felt for their mothers which enabled them to reconcile with

their mothers and see themselves as being part of the community again through the process of forgiveness. This correlates with other findings in the study of childhood abuse where forgiveness of parents was found to have a good impact on emotional health [74]. Forgiveness refers to the process of the voluntary waiving of resentment, hatred, and blame towards others so that not only the person forgets the hostility and hatred but also wishes the best for the transgressor [75]. This forgiveness requires a deliberate act of giving up anger and resentment toward the offender while also not requiring a response from the offender [76, 77]. The offenders, in this case, are the mothers in the communities, and the women succeeded in seeing them as not having a choice in a pressurised community. This is empathy they demonstrated when they put themselves in the shoes of their mothers.

## Strengths and limits of the study

The study is strong in various areas. An important one is the multiplicity of data sources. We used an inductive approach to analyse the data, and the triangulation of different sources of data to enrich the trustworthiness of the findings. Secondary analysis was done from transcripts of 15 women interviewed twice [8], which we compared with the experience of 10 further women with publicly available narratives (see list of books in the methods section). The rich dialogues with experts from different fields, most of whom were not familiar with FGM and the participant of an expert with lived experience also contributed to bringing out concepts that were relevant in enriching the theorisation process as well as in interpreting the data. Unfortunately, due to Covid-19 related restrictions at the time of the study, the experts were not able to meet all together in a presential seminar, which would have had the potential to enrich our findings even further.

The study also presents further limits. First, our emerging theorisation would deserve further testing through a theoretical sampling to ascertain saturation. Moreover, as we were not able to include a sociologist/an anthropologist among our experts as we had initially aimed for our results which attempt to shed light on some complex psychological processes underpinning various stages, may not sufficiently consider the role of social interactions in the change of attitudes towards a social norm. Yet we are aware that social norms are shaped by social interactions and shared cultural values. The modified expectations created by such values and norms in the context of migration might therefore not be sufficiently addressed in our results at this stage. Second, as our proposed model is based exclusively on the narratives of women who have migrated, we do not know about the possible transferability of our result for women who have not left their countries of origin, yet maybe take action to contest FGM in their countries. This would deserve further investigation. Last but not least, the secondary analysis of the original data collected in our previous study [8], might involve some selection bias inherent to our snowball sampling strategy as acknowledged in our previous article.

## Conclusion

Building on hypotheses generated from an earlier study [8], this study examined possible mechanisms which might explain the change in attitudes towards the practice of FGM in migrant women. The model of change which emerged from our analysis gives us an insight into the journey of the women towards changing their attitudes after some self-reported *turning points* in their lives drove them to change. Re-appropriating their own self and identity as well as reconciling themselves with their communities of origin were both valuable in the women's perspectives to move forward with their lives. Trying to protect their daughters from going through the same experience led them to identify FGM as a violation of a woman's right to bodily integrity.

The importance of acknowledging the emotional work and painful traumatic memories awakened during the journey has significant implications for the assisting agencies and health professionals who are in contact with migrant women who originate from countries where FGM is practised. Our model describes several stages including the stage of *The clash*, where the women find themselves trapped in the conflicts of loyalty and which in turn generates very uncomfortable and painful feelings. Thus, our study contributes to the body of knowledge and provides avenues to be explored by health professionals working with migrant women subjected to FGM, to support their processes of reconciliation with themselves and their communities.

## Supporting information

**S1 Appendix. Completed COREQ checklist.**
(DOCX)

**S2 Appendix. A fictive vignette of a typical reconstructed story from different participants after the 1st and 2nd interviews.** (Agboli et al., 2020).
(DOCX)

**S3 Appendix. Interview guide (Agboli et al, 2020).**
(DOCX)

## Acknowledgments

We thank all the women who participated in this study through their narratives, either through their interviews with the first author or through their published life experiences. We would also like to give our special thanks to two '*norm leaders*' and experts with lived experience, Diaryatou Bah and Khady Koita for their valuable inputs at different stages of our study. In addition, we especially thank Dr Olivier Schmitz for his valuable input throughout the beginning of this project.

## Author Contributions

**Conceptualization:** Afi Agboli, Fabienne Richard, Isabelle Aujoulat.

**Formal analysis:** Afi Agboli, Fabienne Richard, Isabelle Aujoulat.

**Investigation:** Afi Agboli, Isabelle Aujoulat.

**Methodology:** Afi Agboli, Isabelle Aujoulat.

**Resources:** Mylene Botbol-Baum, Jean-Luc Brackelaire, Annalisa D'Aguanno, Khadidiatou Diallo, Moïra Mikolajczak, Elise Ricadat.

**Supervision:** Fabienne Richard, Isabelle Aujoulat.

**Writing – original draft:** Afi Agboli.

**Writing – review & editing:** Afi Agboli, Fabienne Richard, Mylene Botbol-Baum, Jean-Luc Brackelaire, Annalisa D'Aguanno, Khadidiatou Diallo, Moïra Mikolajczak, Elise Ricadat, Isabelle Aujoulat.

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
