## [Decision Letter · Decision Letter 0]

17 Dec 2021

PONE-D-21-18726Changing attitudes towards Female Genital Mutilation. From conflicts of loyalty to reconciliation with self and the community: the role of emotion regulation.

PLOS ONE

Dear Dr. Agboli,

Thank you for submitting your manuscript to PLOS ONE. After careful consideration, we feel that it has merit but does not fully meet PLOS ONE’s publication criteria as it currently stands. Therefore, we invite you to submit a revised version of the manuscript that addresses the points raised during the review process.

We invite you to submit a revised version of the manuscript that addresses the points below:

Specifically:

Outdated references indicating changing attitudes toward FGM after migration.

Focusing too much on health consequences especially physical ones in the introduction section rather than explaining attitude change towards FGM.

Not explaining the process of by which experts where selected for interviewing.

We look forward to receiving your revised manuscript.

Kind regards,

Forough Mortazavi

Academic Editor

PLOS ONE

Journal Requirements:

Additional Editor Comments (if provided):

Dear authors,

Thank you for submitting your manuscript to PLOS ONE. The study is relevant and the results are interesting but there is still room for improvement and I hope our comments would help the authors in this regard. We invite you to submit a revised version of the manuscript that addresses the points below:

Is there any model which explains changes in attitudes towards FGM in women who live in their home countries? If so, PLS compare your results with similar studies on women who did not migrate to western countries.

A discussion of potential sources of bias should be included.

Page 7, lines 63-63 needs a reference to other studies.

The methods for recruitment and interviewing the15 women have been stated in your previous paper. No need to restate the ethical considerations presented in your previous article. PLS include a reference to that paper. PLS remove the lines 154-169 and just refer to the original study.

PLS fill out COREQ (COnsolidated criteria for REporting Qualitative research) checklist or other relevant checklists listed by the Equator Network, such as the SRQR, to ensure comprehensive reporting and revise your manuscript if needs be.

Reviewers' comments:

Reviewer's Responses to Questions

**Comments to the Author**

1. Is the manuscript technically sound, and do the data support the conclusions?

Reviewer #1: Yes

Reviewer #2: Yes

2. Has the statistical analysis been performed appropriately and rigorously? 

Reviewer #1: Yes

Reviewer #2: N/A

3. Have the authors made all data underlying the findings in their manuscript fully available?

Reviewer #1: Yes

Reviewer #2: Yes

4. Is the manuscript presented in an intelligible fashion and written in standard English?

Reviewer #1: Yes

Reviewer #2: Yes

5. Review Comments to the Author

Reviewer #1: The study is written in simple (English language with few spelling and grammar mistakes.)

The work is not original but important to the field,

it have the appropriate research work structure, and has an academic vocabulary

The study is complete and in a logical order.

the title is easy to understand and it convey the important aspects of the research

the abstract summarize the article, it include key findings, it introduce the subject and put into perspective ,

Methodology (This study is a A qualitative methodology) every complicated

How it will help >>>

Sample size only 15 small

the research methods and analysis well explained, but difficult to understand.

The study and analysis can get the same outcomes, The analysis is good and appropriate

Plagiarism .. There are some sentences in the manuscript taken from the web.

No conflicts of interests

Ethically approved, the trial was registered

The authors include proper references to previously published methodology,

Tthe description of new methodology is accurate, But need more explanation

The discussion carried out in a satisfactory manner,

T the conclusion support the presented research .

References 61 include proper references to previously published literature cited in the work, but (some not checked and not validated or not checked)

Reviewer #2: The article is very interesting and necessary. I indicate some elements of improvement:

-Is there no bibliographic reference that supports this statement? "The fear of seeing other members of the community performing the practice on their daughters results in women fleeing and migrating to western countries, at times in very difficult conditions, in order to protect their daughters." Example: doi: 10.1186 / s12905-020-00976-w

Neither is the introduction phrase is referenced “Elderly women, guardians of the tradition, always strive to ensure that the practice is passed on from generation to generation”. An example of relevant bibliography in this regard is the following: 10.1186 / s12978-017-0309-2

-The introduction focuses too much on health consequences (especially physical ones) and does not explain what previous studies indicate on attitude change in FGM.

-Among the articles cited on changing attitudes after migration or among migrants living in countries where FGM is not the norm, some are not up-to-date. Nor do you cite some relevant references: https://doi.org/10.1177/1043659620962570

-In the methodology, it would be necessary to explain how the process of choosing the experts to interview was like. Did everyone agree to participate?

-Review outdated bibliography.

-It is necessary to improve figure 1 since some elements are illegible.

6. PLOS authors have the option to publish the peer review history of their article (what does this mean?). If published, this will include your full peer review and any attached files.

Reviewer #1: **Yes: **Prof. Hassan Abduljabbar

Reviewer #2: No

---

## [Author Response · Author response to Decision Letter 0]

26 Mar 2022

To the manuscript Editor: Dr Forough Mortazavi

Academic Editor 

Plos One Afi Agboli

 UCLouvain/IRSS

 30, Clos Chapelle-aux-champs

 1200 Brussels 

Brussels, 28th February 2022

Dear Dr Mortazavi,

Please find attached for your kind review our research revised new manuscript and title for consideration by PLOS ONE.

The title is: Changing attitudes towards female genital mutilation. From conflicts of loyalty to reconciliation with self and the community: The role of emotion regulation 

We would like to thank you for reviewing this article. We have revised the manuscript taking into account the different comments from the reviewers and also from the editors. We have now addressed the transferability issue in the limits section.

We confirm that this work is original and has not been published elsewhere, nor is it currently under consideration for publication elsewhere.

In this paper, we hypothesised that emotion regulation and the resolution of conflicts of loyalty were possible mechanisms that explain the change of attitudes towards the practice of female genital mutilation (FGM). We drew on elements of grounded theory (constant comparison) to investigate how the two mechanisms interacted to explain the change of attitudes of migrant women who were once socialised in the FGM cultural context where the practice is valued and normal, changed their attitudes, and spoke against it.

We believe that our emerging theorisation and the proposed model contribute to the body of knowledge and pave avenues to be explored by health professionals (who work with migrant women with FGM) when they develop a strategic intervention for the prevention of the practice of FGM. 

We do not have any competing interests and we confirm that all authors have approved the manuscript and agree with its submission to PLOS ONE.

We believe that this manuscript is appropriate for publication by PLOS ONE because it gives information to students, academics, and organisations that promote women’s health.

Please address all correspondence concerning this manuscript to us at: 

afi.agboli@uclouvain.be

afisophieagboli@gmail.com

Thank you for your consideration of this manuscript.

Yours sincerely,

Afi Agboli (PhD student).

PS: Please see below the answers to the comments of reviewers.

Answers to reviewers’ comments

PONE-D-21-18726

Changing attitudes towards Female Genital Mutilation. From conflicts of loyalty to reconciliation with self and the community: the role of emotion regulation.

We invite you to submit a revised version of the manuscript that addresses the points below:

Specifically:

Outdated references indicate changing attitudes toward FGM after migration.

Answer: Thank you very much for this comment, which led us to recheck the literature. We have rearranged the introductory part and included more recent references that indicate the change of attitudes towards FGM after migration such as Pastor Bravo et al. (2020); Agboli et al. (2020); Johnsdotter (2018); Johnsdotter (2019); Wahlberg et al. (2018); O’Neill et al., (2018); Walhberg et al. (2017); Johnsdotter & Essen (2016; 2009), Gele et al. (2015), which led us to recheck the literature. However, we still left the former references, which we do not consider as outdated as not much has been published concerning the change of FGM in the context of migration.

The change of attitudes towards FGM in the context of migration (Please see page 3, lines 76-107 of the main manuscript).

Migration context appears to be a favourable platform for the change of views towards the practice of FGM. It leads individuals to reflect on the traditional practices from their countries of origin (Pastor-Bravo et al., 2020). Factors participating in creating an environment that favour the change have been documented. The environmental factors in contexts where the practice is against the law influence the change of attitudes among migrant people (Mackie and LeJeune, 2009; Johnsdotter, 2018; Johnsdotter, 2019). The criminalisation of FGM in Western countries is seen to be an important step towards saving girls from the harmful consequences of FGM (Gele et al., 2015). New environments, the migration journey coupled with anti-FGM interventions, and the encounter with health services have been documented as factors contributing to significant shifts in the women’s attitudes towards FGM as they reject the practice and tradition (Johnsdotter et al., 2009; Lien, 2010; Wahlberg et al., 2018; Agboli et al., 2020; Barrett et al., 2021). Reduced social pressure from extended families was noted because of the distance factor in the context of migration (Johnsdotter et al., 2009), and the length of time spent in host countries (O’Neill et al., 2018; Walhberg et al., 2017; Barrett et al., 2021) also plays a role. Women from the same religious background make them understand that FGM is not a religious principle (Johnsdotter & Essen, 2016).

References:

Pastor-Bravo M del M, Almansa-Martínez P, Jiménez-Ruiz I. Postmigratory Perceptions of Female Genital Mutilation: Qualitative Life History Research. J Transcult Nurs. 2020 Oct 10;1043659620962570. 

Mackie G, LeJeune J. Social dynamics of abandonment of harmful practices: a new look at the theory. Florence, Italy: UNICEF Innocenti Research Centre; 2009. 

Johnsdotter S. Meaning well while doing harm: compulsory genital examinations in Swedish African girls. Sexual and Reproductive Health Matters. 2019;17(2):1586817. 

Gele AA, Sagbakken M, Kumar B. Is female circumcision evolving or dissolving in Norway? A qualitative study on attitudes toward the practice among young Somalis in the Oslo area. Int J Womens Health. 2015;933–43. 

Johnsdotter S, Moussa KM, Carlbom A. “Never My Daughters”: A Qualitative Study Regarding Attitude Change Toward Female Genital Cutting Among Ethiopian and Eritrean Families in Sweden. Health Care For Women International. 2009;114–33. 

Lien I-L, Shultz J-H. Internalizing knowledge and changing attitudes to female genital cutting/mutilation. Obstetrics and Gynecology International. 2013;2013:1–10. 

Barrett HR, Brown K, Alhassan Y, Leye E. Transforming social norms to end FGM in the EU: An evaluation of the REPLACE Approach. Reproductive Health. 2020;17(1). 

Wahlberg A, Johnsdotter S, Selling KE, Kallestal C, Essén B. Baseline data from a planned RCT on attitudes to female genital cutting after migration: when are interventions justified? British Medical Journal [Internet]. 2017; Available from: e017506. doi:10.1136/ bmjopen-2017-017506

Focusing too much on health consequences especially physical ones in the introduction section rather than explaining attitude change towards FGM.

Answer: Thank you very much for this comment. We have now shortened the part on the consequences of FGM to better explain what is already known about the change of attitudes towards FGM after migration and also the challenges on the change in attitudes towards the social norm. Please see the above section and in the main manuscript ‘background section, page 3 lines 66-121.

Not explaining the process by which experts were selected for the interview.

Answer: Thank you for this comment. The process of selection of experts was explained on page 6 lines 140-167. As complex intrapersonal psychological processes were involved in the hypotheses generated by our previous study (Agboli et al., 2020), we aimed at recruiting psychologists with complementary fields of expertise, that we felt were relevant to inform our emerging theory: sexual health and identity issues (Elise Ricadat), emotion competence and regulation (Moïra Mikolajczak), transcultural psychology (Jean-Luc Brackelaire), clinical psychology involved in FGM follow-up and specialised in traumatic memory (Annalisa D’Aguanno). In addition to these various specialties in psychology, we also discussed with a philosopher, specialised in bioethics, norms, and the capabilities approach background (Mylene Botbol-Baum), and an anthropologist who accepted in the first place, but was later unfortunately not available anymore. Last but not least, it was of great importance for us to have the lived experience as an expert who had experienced FGM herself (Kadhiatou Diallo). The experts were approached personally by the first or last author, who knew them for their specific fields of expertise, which did not include FGM-related knowledge or experience for four out of six of them. The first author and the last author discussed the purpose of the study with them and met them individually through videoconference due to the Covid-related restrictions. The consultation with the expert with lived experience was done face-to-face by the first author alone. 

How we approach them: The experts were approached personally by the first author. They were all known either by the supervisor (last author) or by the first author. We would like to stress here that they were very competent in their field of expertise without knowing about FGM except from one who was a member of the steering committee of the first author. The first author and the supervisor discussed the purpose of the study to them, and they agreed in participating in the seminar which unfortunately was not held because of the Covid-19 pandemic.

Answer: Lydia Stephens (now mentioned among the contributors to the article). Please see page 31, line 750.

Is there any model which explains changes in attitudes towards FGM in women who live in their home countries? If so, please compare your results with similar studies on women who did not migrate to western countries.

Answer: Thank you very much for this comment. Indeed, some scholars have studied the change of attitudes towards the social norm in the country of origin and, using existing models of change at the community level have been applied. However, we are not aware of any model of change at the individual level in the country of origin. The Transtheoretical Model (TTM) or stages of change model as developed by Prochaska and DiClemente (1982) has been applied to reflect behaviour change and conceptualised the change of FGM/C in terms of stages of change but the exact identifiable stages could not be defined (Diop et al., 2003; Shell-Duncan, 2006; Toubia and Sharief, 2003). Another model of change at the community level is the social convention theory developed by Mackie and LeJeune (2009) who stressed that as the decision to cut depends on the wider community, a motivated mass of people collectively, deciding to make to abandon the practice publicly might end FGM as it was the case of foot-binding in China. There are challenges in changing attitudes towards FGM and all these approaches and models applied have limitations as far as FGM is concerned, and the impact of interventions aimed at changing attitudes towards FGM was mostly studied at the community level. How changes occur at the individual level remains an under-investigated issue, which this present study aims to address. To date, the different approaches discussed in this section have not led to a reduction of the practice as such due to the insufficiency of strategies that work.

Thus, your point raised is relevant and we have now addressed it as a limit in the strengths and limits section, as we do not know the transferability of our study for women in their country of origin. 

References:

Diop N, Askew A. Strategies for encouraging the abandonment of female genital cutting: experiences from Senegal, Burkina Faso, and Mali. In: Abusharaf RM, editor. Female circumcision: Multicultural perspectives. Philadelphia, USA: University of Pennsylvania Press; 2006. p. 125–41. 

Diop N, Traore F, Diallo H. Study of the effectiveness of training Malian social and health agents in female genital cutting issues and in educating their clients. Final Report. Bamako: Population Council; 2007. https://knowledgecommons.popcouncil.org/departments_sbsr-rh/83/.

Shell-Duncan B, Hernlund Y. Are there ‘stages of change’ in the practice of female genital cutting?: qualitative research fndings from Senegal and The Gambia. African Journal of Reproductive Health. 2006;10(2):57–71. 

Prochaska J O,, Diclemente C O. Satges and processes of sel-change of smoking: toward an integrative model of change. Journal of Consulting and Clinical Psychology. 1983;51(3):390–5. 

Mackie G. Ending footbinding and infibulation: a convention account. American Sociological Review. 1996;61(6):999–1017. 

A discussion of potential sources of bias should be included.

Answer: Thank you for this comment. We have now included it (please see page 29, lines 696-709). 

The study also presents further limits. First, our emerging theorisation would deserve further testing through a theoretical sampling to ascertain saturation. Moreover, as we were not able to include a sociologist/an anthropologist among our experts as we had initially aimed for our results which attempt to shed light on some complex psychological processes underpinning various stages, may not sufficiently consider the role of social interactions in the change of attitudes towards a social norm. Yet we are aware that social norms are shaped by social interactions and shared cultural values. The modified expectations created by such values and norms in the context of migration might therefore not be sufficiently addressed in our results at this stage. Moreover, as our proposed model is based exclusively on the narratives of women who have migrated, we do not know about the possible transferability of our result for women who have not left their countries of origin, yet maybe take action to contest FGM in their countries. This would deserve further investigation. Last but not least, the secondary analysis of the original data collected in our previous study (Agboli et al., 2020), might involve some selection bias inherent to our snowball sampling strategy as acknowledged in our previous article. 

Page 7, lines 61-63 needs a reference to other studies. Consequently, the fear of seeing other members of the community performing the practice on their daughters results in women fleeing and migrating to western countries, at times in very difficult conditions, in order to protect their daughters.

Answer: Thank you very much for this comment. We have now added other references related to the statement. Please see page 3, line 63-65 in the main manuscript. The references are below:

Alradie-Mohamed A, Kabir R, Arafat Y. Decision-Making Process in Female Genital Mutilation: A Systematic Review. Int J Environ Res Public Health. 2020;3362. 

Agboli A. Overcoming female genital mutilation and internalised social norms in the context of migration: Turning points and explanatory mechanisms. Doctoral thesis, Université Catholique de Louvain; 2021. Available from: http://hdl.handle.net/2078.1/250644

Agboli A.A., Richard F., Aujoulat I. ‘When my mother called me to say that the time of cutting had arrived, I just escaped to Belgium with my daughter’: identifying turning points in the change of attitudes towards the practice of female genital mutilation among migrant women in Belgium. BMC Women’s Health. 2020;20:107.

The methods for recruitment and interviewing the15 women have been stated in your previous paper. No need to restate the ethical considerations presented in your previous article. PLS include a reference to that paper. PLS remove the lines 154-169 and just refer to the original study.

Answer: Thank you for this comment. We have now provided the reference and removed lines 154-169 as instructed.

PLS fill out COREQ (COnsolidated criteria for REporting Qualitative research) checklist or other relevant checklists listed by the Equator Network, such as the SRQR, to ensure comprehensive reporting and revise your manuscript if needed be.

Answer: Thank you. Completed form uploaded as supporting file.

Completed COREQ Checklist_Plos

No Item Guide Q/ description

Domain 1

Research team and 

reflexivity

Personal 

characteristic

1. Interviewer/facilitator The first author conducted the interviews

2. Credentials BSc, MSc

3. Occupation PhD student, Registered Nurse

4. Gender Female

5. Experience and training Registered Nurse, PhD Study in Public health

Relationship with

 participants

6. Relationship established Yes, through gatekeepers and snowball procedures.

7. Participant knowledge of the interviewer No prior knowledge of the participants except the expert Kadyatou Diallo, the president of GAMS (Groupe pour l’Abolition des Mutilations Sexuelle) who later became involved in the theorisation process.

8. Interviewer characteristics The interviewer is an African woman, a PhD student who is also the first author, and a health professional (nursing background). 

Domain 2

Study design

Theoretical 

Framework

9. Methodological orientation and Theory A qualitative methodology informed by the narrative biographical approach and grounded theory approach was used to allow the theorisation process. Page 5, lines 124-130.

Participant selection

10. Sampling 3 data sources: see page 6-7, lines 131-170

-the 15 women from the previous study were interviewed twice recruited through gatekeepers and snowball procedures.

- 10 books: the life stories and public testimonies of 10 women referred to as ‘norm leaders’ through websites and contact with organisation very active in the field.

- 6 experts: chosen for their complementary fields of expertise. Page 7

11. Method of approach -the 15 women were recruited through gatekeepers and snowball procedures.

- 10 books: the life stories and public testimonies of 10 women referred to as ‘norm leaders’ through websites and contact with the organisation very active in the field.

- 6 experts: personally approached for their theoretical expertise with regards to the emerging hypotheses and with no prior knowledge of FGM for 4 of them. One expert with lived experience of FGM, and one with clinical experience in relation to FGM. They were recruited in the University (UCLouvain, Belgium) and the one expert with lived experience is the founder and president of GAMS and was recruited in the organisation. Page 7,lines 131-170.

12. Sample size -30 transcripts from 15 women interviewed twice.(previous study 1)

-10 books Page 6, line 138

-6 experts Page 6

13. Non-participation N/A

Setting

14. Setting of data collection -15 women were interviewed either at GAMS, their homes, or in the office of the first author according to their preferences (Agboli et al., 2020) (Agboli A.A., Richard F., Aujoulat I. ‘When my mother called me to say that the time of cutting had arrived, I just escaped to Belgium with my daughter’: identifying turning points in the change of attitudes towards the practice of female genital mutilation among migrant women in Belgium. BMC Women’s Health. 2020;20: 107).

-the books

-The experts were interviewed via videoconference through Teams by the first and last authors. The consultation with the lived experience expert was done face-to-face by the first author alone. Page 6-7, lines 131-170.

15. Presence of non-participants N/A

16. Description of sample 

-The sample of the 15 women is described in the article (Agboli A.A., Richard F., Aujoulat I. ‘When my mother called me to say that the time of cutting had arrived, I just escaped to Belgium with my daughter’: identifying turning points in the change of attitudes towards the practice of female genital mutilation among migrant women in Belgium. BMC Women’s Health. 2020;20:107).

-the books, see page 6-7.

-The experts: Page 6-7 

Data collection

17. Interview guide -The interview guide of the 15 women is in article (Agboli et al., 2020).

-The interviews of the experts were based on emerging hypotheses from our study 1 (Agboli et al., 2020)

18. Repeat interviews Repeat interviews of the 15 women. See article (Agboli et al., 2020)

19. Audio/ visual recording Audio recording: The women’s stories were recorded, and their consent was asked for beforehand in the previous study.(see Agboli et al., 2020).

20. Field notes Yes. Extensive notes were taken during the interviews with the experts and we sent the synthesis which they had the opportunity to read, revise and validate. Page 7, lines 159-167.

21. Duration -The 15 women: see Agboli et al., 2020

-Consultation with experts: between 45 and 1h. The duration of the face-to-face interview was 45 min.

Page 7, lines 156-167.

22. Data saturation Our emerging theorisation would deserve further testing through theoretical sampling to ascertain saturation. Page 28 lines 698-711.

The multiplicity of data sources: the original interviews, the books analysis and the consultation with the experts tend towards saturation even though we are aware that our model could be further tested through theoretical sampling. Page 28-29 lines 684-709.

23. Transcripts returned Yes.

-the 15 women: had a chance to look at the lifelines and co-constructed the turning points with us, confirmed them, and validated them (see Agboli et al. 2020).

-The experts: synthesis of notes taken returned to them and validated them. Page 7, lines 163-167.

Domain 3

Analysis and findings

Data analysis

24. Number of data coders Up to three: 

-The 15 women: Three researchers (AA, IA, FR) were involved in the analysis and interpretation of the 30 transcripts.

-The analysis of the books: 2nd author (FR) and the first author (AA).

-The experts: Analysis and theoretical integration of the experts’ contributions were done by the first author (AA) and the last author (IA).

See page 10-11, lines 238-268.

25. Description of the coding tree Yes, the authors provided a description of the initial model of emotion regulation and the conflict of loyalty. 

26. Derivation of themes We started analysing the data using a framework analysis approach based on the hypotheses generated from the previous study (Agboli et al., 2020) and proceeded inductively with predefined categories, and developed our model of change. See Page 11, lines 261-268.

27. Software We used Excel software to group the codes.

28. Participant checking Yes.

-The 15 women: the second interview and co-construction of turning points (Agboli et al., 2020)

-The experts: revision and validation of the synthesis of their interviews sent to them and later, the discussion, interpretation, and validation of the final model. Page 11; lines 246-268.

Reporting

29. Quotations presented Yes, pages 12-22.

30. Data and findings consistent Yes, pages 12-22

31. Clarity of major themes Yes, page 12-22.

32. Clarity of minor themes N/A

Reviewer #1: The study is written in simple (English language with few spelling and grammar mistakes.) The work is not original but important to the field, it has the appropriate research work structure, and has an academic vocabulary

The study is complete and in a logical order. the title is easy to understand and it conveys the important aspects of the research. the abstract summarizes the article, it includes key findings, it introduces the subject and put into perspective,

Methodology (This study is a qualitative methodology) every complicated

How it will help >>>

Sample size only 15 small

Answer: Thank you for this comment. 

The sample size of 15 women is possibly small to reach saturation regarding our theorisation process however compensated through the triangulation of the data sources such as the 10 published narratives and the consultations with the experts. Please see page 5, line 123 (Methods section).

This sample size was enough for the first part of our study (Agboli et al., 2020) which aimed at identifying turning points where the women participants were interviewed twice.

Furthermore, the concept of saturation is tied to the methodology applied as far as qualitative studies are concerned. A study done by Malterud et al. (2016) asserts that fewer participants may be required when the communication between researcher and participants is strong (pp.1755). We believe that our research has a strength when the findings of the first study which used the life story of 15 women, were co-constructed by the interaction between the researcher and the participants. 

Reference

Agboli A, Richard F, Aujoulat I. "When my mother called me to say that the time of cutting had arrived, I just escaped to Belgium with my daughter": identifying turning points in the change of attitudes towards the practice of female genital mutilation among migrant women in Belgium. BMC Women’s Health. 2020; 20(1):107. doi:10.1186/s12905-020-00976-w.(pp9)

Malterud K, Siersma V D, Guassara AD. Sample size in qualitative interview studies: guided by information power. Qualitative Health Research. 2016; 26(3): 1753-1760

Doi: 10.1177/1049732315617444

The research methods and analysis are well explained, but difficult to understand.

The study and analysis can get the same outcomes, The analysis is good and appropriate

Plagiarism .. There are some sentences in the manuscript taken from the web.

Answer: Thank you for this comment. We are not aware of any plagiarism issue and we screened our article through UCLouvain plagiarism software (Anti-plagiarism Compilatio) and found only 5% of similarity.

No conflicts of interests

Ethically approved, the trial was registered

The authors include proper references to previously published methodology,

The description of the new methodology is accurate, But need more explanation

The discussion was carried out in a satisfactory manner,

The conclusion supports the presented research.

References 61 include proper references to previously published literature cited in the work, but (some not checked and not validated or not checked)

Answer: Thank you very much for these comments. We have now corrected reference 61 which has become reference 75:

Menahem S and Love M. Forgiveness in psychotherapy: The key to healing. J Clin Psychol. 2013;829–35.

Reviewer #2: The article is very interesting and necessary. I indicate some elements of improvement:

-Is there no bibliographic reference that supports this statement? "The fear of seeing other members of the community performing the practice on their daughters results in women fleeing and migrating to western countries, at times in very difficult conditions, in order to protect their daughters." Example: doi: 10.1186 / s12905-020-00976-w

Answer: Thank you very much for this comment. We have now added references related to the statement. 

Agboli A. Overcoming female genital mutilation and internalised social norms in the context of migration: Turning points and explanatory mechanisms. Doctoral Thesis, Université Catholique de Louvain; 2021. Available from: http://hdl.handle.net/2078.1/250644

Alradie-Mohamed A, Kabir R, Arafat YSM. Decision-Making Process in Female Genital Mutilation: A Systematic Review. Int J Environ Res Public Health. 2020; 17(10): 3362.

doi: 10.3390/ijerph17103362

Neither is the introductory phrase is referenced “Elderly women, guardians of the tradition, always strive to ensure that the practice is passed on from generation to generation”. An example of relevant bibliography in this regard is the following: 10.1186 / s12978-017-0309-2

Answer: Thank you for this relevant comment. We have now added the references. Please see page 3, lines 61-63.

Koukoui S, Hassan G, Guzder J. The mothering experience of women with FGM/C raising ‘uncut’ daughters, in Ivory Coast and in Canada. Reprod Health. 2017; 14(1):51. 

https://doi.org/10.1186/s12978-017-0309-2

Shell-Duncan B, Moreau A, Wander K, Smith S. The role of older women in contesting norms associated with female genital mutilation/cutting in Senegambia: a factorial focus group analysis. PloS One; 13(7):e0199217

-The introduction focuses too much on health consequences (especially physical ones) and does not explain what previous studies indicate on attitude change in FGM.

Answer: Thank you very much for this comment. We have answered this comment above, page 1 of this file. Please see page 3, lines 66-121 of the main manuscript.

-Among the articles cited on changing attitudes after migration or among migrants living in countries where FGM is not the norm, some are not up-to-date. Nor do you cite some relevant references: https://doi.org/10.1177/1043659620962570

Answer: Thank you very much for this comment. Please see above, previous comments. Page 3, lines 57-121 on the main manuscript.

-In the methodology, it would be necessary to explain how the process of choosing the experts to interview was like. Did everyone agree to participate?

Answer: Thank you for this comment. The process of selection of the experts is now better explained on page 6 lines 140-167. 

As complex intrapersonal psychological processes were involved in the hypotheses generated by our previous study (Agboli et al., 2020), we aimed at recruiting psychologists with complementary fields of expertise, that we felt were relevant to inform our emerging theory: sexual health and identity issues (Elise Ricadat), emotion competence and regulation (Moïra Mikolajczak), transcultural psychology (Jean-Luc Brackelaire), clinical psychology involved in FGM follow-up and specialised in traumatic memory (Annalisa D’Aguanno). In addition to these various specialties in psychology, we also discussed with a philosopher, specialised in bioethics, norms, and the capabilities approach background (Mylene Botbol-Baum), and an anthropologist who accepted in the first place, but was later unfortunately not available anymore. Last but not least, it was of great importance for us to have the lived experience as an expert who had experienced FGM herself (Kadhiatou Diallo). The experts were approached personally by the first or last author, who knew them for their specific fields of expertise, which did not include FGM-related knowledge or experience for four out of six of them. The first author and the last author discussed the purpose of the study with them and met them individually through videoconference due to the Covid-related restrictions. The consultation with the expert with lived experience was done face-to-face by the first author alone. 

How we approach them: The experts were approached personally by the first author. They were all known either by the supervisor (last author) or by the first author. We would like to stress here that they were very competent in their field of expertise without knowing about FGM except from one who was a member of the steering committee of the first author. The first author and the supervisor discussed the purpose of the study to them, and they agreed in participating in the seminar which unfortunately was not held because of the Covid-19 pandemic.

-Review outdated bibliography.

Answer: Thank you for this comment. We do not think that the background references are outdated as such. However, we improved the section on the change of attitudes after migration with more recent references.

-It is necessary to improve figure 1 since some elements are illegible.

Answer: Thank you for this comment. We have rearranged Fig1 as instructed. Please see below.

---

## [Decision Letter · Decision Letter 1]

8 Apr 2022

PONE-D-21-18726R1Changing attitudes towards Female Genital Mutilation. From conflicts of loyalty to reconciliation with self and the community: the role of emotion regulation.PLOS ONE

Dear Dr. Agboli,

Thank you for submitting your manuscript to PLOS ONE. After careful consideration, we feel that it has merit but does not fully meet PLOS ONE’s publication criteria as it currently stands. Therefore, we invite you to submit a revised version of the manuscript that addresses the points raised during the review process.

We look forward to receiving your revised manuscript.

Kind regards,

Forough Mortazavi

Academic Editor

PLOS ONE

Journal Requirements:

Reviewers' comments:

Reviewer's Responses to Questions

**Comments to the Author**

1. If the authors have adequately addressed your comments raised in a previous round of review and you feel that this manuscript is now acceptable for publication, you may indicate that here to bypass the “Comments to the Author” section, enter your conflict of interest statement in the “Confidential to Editor” section, and submit your "Accept" recommendation.

Reviewer #1: All comments have been addressed

Reviewer #2: All comments have been addressed

2. Is the manuscript technically sound, and do the data support the conclusions?

Reviewer #1: Yes

Reviewer #2: Yes

3. Has the statistical analysis been performed appropriately and rigorously? 

Reviewer #1: Yes

Reviewer #2: N/A

4. Have the authors made all data underlying the findings in their manuscript fully available?

Reviewer #1: Yes

Reviewer #2: No

5. Is the manuscript presented in an intelligible fashion and written in standard English?

Reviewer #1: Yes

Reviewer #2: Yes

6. Review Comments to the Author

Reviewer #1: thank you for your comments

The study is written in simple (English language with few spelling and grammar mistakes.)

It have the appropriate research work structure, and has an academic vocabulary

The study is complete and in a logical order.

the title is easy to understand and it convey the important aspects of the research

the abstract summarize the article, it include key findings, it introduce the subject and put into perspective ,

the research methods and analysis well explained.

The study and analysis can get the same outcomes, The analysis is good and appropriate

No conflicts of interests

Ethically approved.

The authors include proper references to previously published methodology,

The description of new methodology is accurate,

The discussion carried out in a satisfactory manner,

T the conclusion support the presented research .

References

Accept

Reviewer #2: Congratulation. The manuscript has improved significantly. I only have a couple of comments for improvements.

-In the method section review the next phrase:

A qualitative methodology informed by the grounded theory approach was used to allow us and

125 it allowed us to generate the hypotheses mentioned above [8].

- It could be confusing to readers that there are 30 transcripts from 15 participants. At some point it should be explained that 2 interviews were conducted with each participant and the reason.

- In data source section:

As in the first data source is detailed. It should also be detailed in the third data source that the transcripts of the interviews, field notes... are the data to be analyzed.

7. PLOS authors have the option to publish the peer review history of their article (what does this mean?). If published, this will include your full peer review and any attached files.

Reviewer #1: **Yes: **Prof. HassaN Abduljabbar

Reviewer #2: No

---

## [Author Response · Author response to Decision Letter 1]

19 May 2022

To the manuscript Editor: Dr Forough Mortazavi

Academic Editor 

Plos One Afi Agboli

 UCLouvain/IRSS

 30, Clos Chapelle-aux-champs

 1200 Brussels 

Brussels, 09th May 2022

Dear Dr Mortazavi,

Please find attached for your kind review our revised new manuscript and title for consideration by PLOS ONE.

The title is: Changing attitudes towards female genital mutilation. From conflicts of loyalty to reconciliation with self and the community: The role of emotion regulation 

We would like to thank you for reviewing this article. We have revised the manuscript taking into account the different comments from the reviewers and also from the editors. We added one reference as requested by Reviewer 2.

We confirm that this work is original and has not been published elsewhere, nor is it currently under consideration for publication elsewhere.

In this paper, we hypothesised that emotion regulation and the resolution of conflicts of loyalty were possible mechanisms that explain the change of attitudes towards the practice of female genital mutilation (FGM). 

We believe that our emerging theorisation and the proposed model contribute to the body of knowledge and pave avenues to be explored by health professionals (who work with migrant women with FGM) when they develop a strategic intervention for the prevention of the practice of FGM. 

We do not have any competing interests and we confirm that all authors have approved the manuscript and agree with its submission to PLOS ONE.

We believe that this manuscript is appropriate for publication by PLOS ONE because it gives information to students, academics, and organisations that promote women’s health.

Please address all correspondence concerning this manuscript to us at: 

afi.agboli@uclouvain.be

afisophieagboli@gmail.com

Thank you for your consideration of this manuscript.

Yours sincerely,

Afi Agboli (PhD student).

PS: Please see below the answers to the comments of reviewers.

Revision 2

Reviewer #2: Congratulation. The manuscript has improved significantly. I only have a couple of comments for improvements.

-In the method section review the next phrase: A qualitative methodology informed by the grounded theory approach was used to allow us and 125 it allowed us to generate the hypotheses mentioned above [8].

Answer: Thank you very much for this comment. We have now rearranged the sentence. Please see Page 5 lines 124-125 in the main manuscript.

 A qualitative methodology informed by the grounded theory approach was used to allow us to generate the hypotheses mentioned above [8].

- It could be confusing to readers that there are 30 transcripts from 15 participants. At some point, it should be explained that 2 interviews were conducted with each participant and the reason.

Answer: Thank you very much for this comment. We have corrected it. Please see page 6, line 134-144 of the main manuscript.

(a) Fifteen women were interviewed twice in our previous study [8]. We chose the method of biographical narrative interviewing method (BNIM) as developed by Wengraf [35] in order to produce narratives relating to life events. In the BNIM approach to data collection, the interviewee is seen in two phases and sometimes three (not always present), with the first interview being unstructured and the consecutive interviews building on the previously collected data [35]. Thus, we had 30 transcripts derived from the interviews of the 15 women participants in this study who had all undergone FGM in their countries of origin during childhood. They all originated from sub-Saharan African countries and were living in Belgium at the time of the interview. Full details of the sample and the description of the process of the interviews can be found in our publication [8]. 

- In data source section:

As in the first data source is detailed. It should also be detailed in the third data source that the transcripts of the interviews, field notes... are the data to be analyzed.

Answer: Thank you very much for the comment. We highlighted it on page 11, lines 257-260 in the main manuscript.

The notes taken during the different discussions were compared and analysed by the first and last authors to further clarify our emerging theorisation, which was also discussed with co-author FR on several occasions and enriched through a review of some published material suggested by the experts.

---

## [Decision Letter · Decision Letter 2]

6 Jun 2022

Changing attitudes towards Female Genital Mutilation. From conflicts of loyalty to reconciliation with self and the community: the role of emotion regulation.

PONE-D-21-18726R2

Dear Dr. Agboli,

We’re pleased to inform you that your manuscript has been judged scientifically suitable for publication and will be formally accepted for publication once it meets all outstanding technical requirements.

Kind regards,

Forough Mortazavi

Academic Editor

PLOS ONE

Additional Editor Comments (optional):

Reviewers' comments:

Reviewer's Responses to Questions

**Comments to the Author**

1. If the authors have adequately addressed your comments raised in a previous round of review and you feel that this manuscript is now acceptable for publication, you may indicate that here to bypass the “Comments to the Author” section, enter your conflict of interest statement in the “Confidential to Editor” section, and submit your "Accept" recommendation.

Reviewer #1: All comments have been addressed

2. Is the manuscript technically sound, and do the data support the conclusions?

Reviewer #1: Yes

3. Has the statistical analysis been performed appropriately and rigorously? 

Reviewer #1: Yes

4. Have the authors made all data underlying the findings in their manuscript fully available?

Reviewer #1: Yes

5. Is the manuscript presented in an intelligible fashion and written in standard English?

Reviewer #1: Yes

6. Review Comments to the Author

Reviewer #1: The study is written in simple (English language with few spelling and grammar mistakes.)

It have the appropriate research work structure, and has an academic vocabulary

The study is complete and in a logical order.

the title is easy to understand and it convey the important aspects of the research

the abstract summarize the article, it include key findings, it introduce the subject and put into perspective ,

the research methods and analysis well explained.

The study and analysis can get the same outcomes, The analysis is good and appropriate

No conflicts of interests

Ethically approved.

The authors include proper references to previously published methodology,

The description of new methodology is accurate,

The discussion carried out in a satisfactory manner,

T the conclusion support the presented research .

References

7. PLOS authors have the option to publish the peer review history of their article (what does this mean?). If published, this will include your full peer review and any attached files.

Reviewer #1: **Yes: **Prof. Hassan Abduljabbar

---

## [Editor Report · Acceptance letter]

10 Jun 2022

PONE-D-21-18726R2 

Changing attitudes towards Female Genital Mutilation. From conflicts of loyalty to reconciliation with self and the community: the role of emotion regulation. 

Dear Dr. Agboli:

I'm pleased to inform you that your manuscript has been deemed suitable for publication in PLOS ONE. Congratulations! Your manuscript is now with our production department. 

Kind regards, 

on behalf of

Dr. Forough Mortazavi 

Academic Editor

PLOS ONE